

# Improving the calibration-free complementary evaporation principle by linking with the Budyko framework

Daeha Kim[1], Minha Choi[2], Jong Ahn Chun[3]

[1]Department of Civil Engineering, Jeonbuk National University, Jeonju, Jeollabuk-do, 54896, South Korea
[2]Department of Water Resources, Sungkyunkwan University, Suwon, Gyeonggi-do, 16419, South Korea
[3]Prediction Research Department, APEC Climate Center, Busan, 48058, South Korea

*Correspondence to*: Jong Ahn Chun (jachun@apcc21.org)

**Abstract.** While it has performed well in predicting terrestrial evapotranspiration ($ET_a$) in many gauged locations over the
world, the calibration-free complementary relationship (CR) depends on a questionable assumption that the Priestley-Taylor
coefficient ($\alpha_e$) is spatially constant over an extensive area. In this work, we evaluated the predictive performance of this
convenient method, which only requires atmospheric inputs, against in-situ flux observations and water balance estimates
($ET_{wb}$) in Australia. We found that the CR method with a spatially constant $\alpha_e$ derived from fractional wet areas did not
perform as highly as previous studies would suggest, underperforming three advanced $ET_a$ models in closing basin-scale
water balance. This problem was remedied by linking the CR method with a traditional Budyko equation that allowed
upscaling of optimal $\alpha_e$ values from gauged basins to ungauged locations. The CR method with the $\alpha_e$ upscaled by the
atmospheric inputs and the mean precipitation (P) better reproduced the grid $ET_{wb}$ available over the entire continent, and
outperformed the three $ET_a$ models. This study suggests that the fixed $\alpha_e$ could lead the CR method to biased $ET_a$ estimates,
and it needs to be constrained by climate conditions to better close local water budgets.

**1 Introduction**

Evapotranspiration ($ET_a$) links water and energy exchanges between lands and the atmosphere. On the global scale, more
than 60% of terrestrial precipitation (P) returns to the atmosphere through plants' vascular systems and soil pores, while
consuming over 70% of surface net radiation (Trenberth et al., 2009; 2007). Since it is tightly coupled with carbon cycles,
abnormally low $ET_a$ indicates food insecurity and low ecosystem sustainability (Pareek et al., 2020; Kyatengerwa et al.,
2020; Jasechko, 2018; Swann et al., 2016). In severe cases, $ET_a$ limited by soil moisture can lead to extreme heatwaves that
further propagate the water deficit in space and time (Schumacher et al., 2022; Miralles et al., 2014; Mueller and
Seneviratne, 2012).

Despite great community efforts for sharing in-situ observations (e.g., Baldocchi, 2020; Novick et al., 2018),
gauging networks for $ET_a$ are still unevenly established over the world and often subject to limited data lengths (Ma et al.,
2021). Unavoidably, modeling approaches are needed to predict $ET_a$ in ungauged or poorly gauged locations, or to



characterize it at larger spatial and longer temporal scales. A wide range of modeling frameworks have been proposed such as physical models (e.g., Martens et al., 2017; Zhang et al., 2016), machine-learning techniques (e.g., Jung et al., 2019; Tramontana et al., 2016), and conceptual land surface schemes (e.g., Guimberteau et al., 2018; Haverd et al., 2018).

However, most of the $ET_a$ models require P data and/or land surface information (e.g., remote-sensing vegetation indices) as major inputs. Owing to high uncertainty associated with P data (Sun et al., 2018) and model structure and parameterization (Zhang et al., 2019; Samaniego et al., 2017), $ET_a$ models have produced substantial disparity in their estimates. In the comprehensive intercomparison by Pan et al. (2020), for example, the spread of the global mean $ET_a$ simulated by 14 land surface schemes was larger than 200 mm $a^{-1}$, and similar incongruity between modeled $ET_a$ estimates had found in the earlier Global Soil Wetness Project (Schlosser and Gao, 2010). This suggests a necessity of an alternative

method to circumvent the use of P and synthesized soil moisture.

A practical method to simulate $ET_a$ without P and land-surface information is the complementary relationship (CR) of evaporation (Bouchet, 1963). It uses the evident fact that the air over a water-limited surface amplifies its vapor pressure deficit (VPD), while this effect disappears when the same surface is amply wet (Zhou et al., 2019; Chen and Buchberger, 2018; Ramírez et al., 2005). This atmospheric self-adjustment could become a predictor of water-limited $ET_a$, and various

methods have been formulated (e.g., Anayah and Kaluarachchi, 2014; Crago and Qualls, 2013; Huntington et al., 2011; Kahler and Brutsaert, 2006; Crago and Crowley, 2005; and Hobbins et al., 2004 among others). In particular, the non-dimensional derivation of Brutsaert (2015) and following modifications (Crago and Qualls, 2021; Szilagyi, 2021; Szilagyi et al., 2017; Crago et al., 2016) provided the generality and thermodynamic foundations of Bouchet's (1963) principle.

The non-dimensional CRs derived from definitive boundary conditions have showed outstanding performance in

reproducing $ET_a$ observations at local, regional, and global scales (e.g., Ma et al., 2021; Brutsaert et al., 2020; Ma and Szilagyi, 2019; Ma et al., 2019; Crago and Qualls, 2018; Brutsaert et al., 2017), and their applications have extended to drought risk assessments (e.g., Kim et al., 2021; Kyatengerwa et al., 2020; Kim et al., 2019). However, to date, the only formulation that purely requires meteorological data and thus usable in ungauged areas is the one by Szilagyi et al. (2017). The other kindred methods depend on any reference $ET_a$ data (e.g., eddy-covariance flux data or water-balance estimates) to

calibrate associated parameters that determine the hypothetical wet-environment evapotranspiration ($ET_w$). To resolve this problem, Szilagyi et al. (2017) analytically estimated the Priestley-Taylor coefficients ($\alpha_e$) in wet locations only using atmospheric observations, and transferred their average value to the entire area of interest. This convenient calibration-free approach had well closed basin-scale water balance in the conterminous U.S. (Ma and Szilagyi, 2019), China (Ma et al., 2019), and 52 major river basins over the world (Ma et al., 2021).

Nonetheless, it seems to be an oversimplification to assume that $\alpha_e$ is constant over a large continental area. On many open-water surfaces, $\alpha_e$ has varied substantially on sub-daily, daily, monthly, and annual timescales (Han et al., 2021; Assouline et al., 2016; Baldocchi et al., 2016; Wang et al., 2014; Parlange and Katul, 1992). Given the space-time links between climate, soil, and vegetation (Hagedorn et al., 2019; Mekonnen et al., 2019; Rodriguez-Iturbe, 2000), the aerodynamic component of $ET_w$ may not be described simply by a fixed fraction of the surface net radiation. The constant $\alpha_e$





assumption might be unable to close surface energy balance under diverse climate and surface conditions, because the aerodynamic resistance plays a pivotal role in modulating surface temperature (Chen et al., 2020).

In this work, we proposed how to mend the problematic assumption of constant $\alpha_e$ in the Australian continent, where performance of the calibration-free CR has not yet thoroughly evaluated. By linking the CR and a traditional Budyko framework, here we analytically addressed why $\alpha_e$ cannot be fully independent of local climate conditions, and how it can be

upscaled from gauged to ungauged locations while being constrained by local climate conditions.

## 2 Methodology and data

### 2.1 Calibration-free CR formulation by Szilagyi et al. (2017)

The CR of Szilagyi et al. (2017) describes the self-adjustment of $ET_p$ to surface moisture conditions using three evaporation rates, namely, $ET_a$, $ET_w$, and the atmospheric evaporative potential ($ET_p$). Again, $ET_a$ is the actual moisture flux from a land

surface to the atmosphere, and $ET_w$ is the hypothetical $ET_a$ rate that should occur with ample water availability. $ET_p$ is the atmospheric capacity to receive water vapor that responds actively to soil moisture conditions. By defining the two dimensionless variables as $x \equiv ET_w/ET_p$ and $y \equiv ET_a/ET_p$, and a definitive relationship between x and y could be derived from four boundary conditions.

Under ample water conditions, $ET_p$ does not deviate from $ET_w$ and $ET_a$ (i.e., $ET_p = ET_w = ET_a$); hence, the

corresponding zero-order boundary condition is (i) y = 1 for x = 1. In contrast, $ET_a$ must be nil over a desiccated surface (i.e., y = 0), and by energy balance, the surface net radiation should be fully transformed to the sensible heat flux. Then, the atmospheric VPD would be amplified at the maximum level under the given radiative forcing. Defining the maximum $ET_p$ rate as $E_{pmax}$, another zero-order boundary condition is given as (ii) y = 0 for $x = x_{min} \equiv ET_w/E_{pmax}$. When x = 1 (i.e., ample water conditions), changes in $ET_a$ would be controlled by changes in $ET_w$, yielding a first-order boundary condition as: (iii)

dy/dx = 1 for x = 1. Over a desiccated surface, the zero $ET_a$ cannot change irrespective of changes in $ET_w$ and $ET_p$; thus, another first-order boundary condition becomes (iv) dy/dx = 0 for x = 0. The simplest polynomial equation satisfying the four boundary conditions is:

$$y = 2X^2 - X^3, \tag{1a}$$

where, X rescales the dimensionless x into [0, 1] as:

$$X = \frac{x - x_{min}}{1 - x_{min}} = \frac{E_{pmax} - ET_p}{E_{pmax} - ET_w}\frac{ET_w}{ET_p}. \tag{1b}$$

Eq. (1) allows users to estimate $ET_a$ with no land-surface information, because $ET_p$, $ET_w$, and $E_{pmax}$ are all obtainable from a set of net radiation, air temperature, dew-point temperature, and wind speed data. $ET_p$ and $E_{pmax}$ could be estimated by the Penman (1948) equation:

$$ET_p = \frac{\Delta(T_a)}{\Delta(T_a) + \gamma}\frac{R_n}{\lambda_v} + \frac{\gamma}{\Delta(T_a) + \gamma}f_u VPD, \tag{2}$$






$$E_{pmax} = \frac{\Delta(T_{dry})}{\Delta(T_{dry})+\gamma}\frac{R_n}{\lambda_v} + \frac{\gamma}{\Delta(T_{dry})+\gamma}f_u e_s(T_{dry}),$$ (3)

where, $\Delta(T)$ is the slope of the saturation vapor pressure curve (kPa °C$^{-1}$) at a temperature T, $T_a$ is the mean air temperature (°C), $\gamma$ is the psychometric constant (kPa °C$^{-1}$), $R_n$ is the surface net radiation less the soil heat flux (MJ m$^{-2}$ d$^{-1}$), $\lambda_v$ is the latent heat of vaporization (MJ kg$^{-1}$), $f_u = 2.6 (1 + 0.54 u_2)$ is the Rome wind function (mm d$^{-1}$ kPa$^{-1}$), where $u_2$ is the 2-m wind speed (m s$^{-1}$), and VPD is $e_s(T_a)$ minus $e_s(T_{dew})$, where $e_s(T)$ is the saturation vapor pressure at T and $T_{dew}$ is the dew

point temperature (°C).

$T_{dry}$ in Eq. (3) is the air temperature (°C) at which the boundary layer is devoid of humidity by the adiabatic drying process:

$$T_{dry} = T_{wb} + \frac{e_s(T_{wb})}{\gamma} = T_a + \frac{e_s(T_{dew})}{\gamma},$$ (4)

where, $T_{wb}$ is the wet-bulb temperature (°C), where the saturation vapor pressure curve intersects with the adiabatic wetting

line:

$$\gamma\frac{T_{wb}-T_{avg}}{e_s(T_{wb})-e_a} = -1.$$ (5)

To quantify $ET_w$, the Priestly and Taylor (1972) equation has been a typical choice (e.g., Han and Tian, 2018; Szilagyi et al., 2017; Crago et al., 2016; Brutsaert, 2015):

$$ET_w = \alpha_e \frac{\Delta(T_w)}{\Delta(T_w)+\gamma}\frac{R_n}{\lambda_v},$$ (6)

where, $\alpha_e$ varies usually within [1.10, 1.32] (Szilagyi et al., 2017), and $T_w$ is the wet-environment air temperature (°C). $T_w$ can be approximated with the wet-surface temperature ($T_{ws}$), because negligible vertical air temperature gradient is observable in wet environments. Given the independence of $T_{ws}$ on areal extent (Szilagyi and Schepers, 2014), it is obtainable by iteration from the Bowen ratio ($\beta$) of a small wet patch:

$$\beta = \frac{R_n-ET_p}{ET_p} \approx \gamma \frac{T_{ws}-T_a}{e_s(T_{ws})-e_s(T_{dew})}.$$ (7)

The approximate Eq. (7) assumes that the available radiation for the wet patch is close to that of the drying surface (Szilagyi et al., 2017). $T_{ws}$ might be higher than $T_a$ when the air close to saturation. In such a case, $T_{ws}$ needs to be constrained by $T_a$ when estimating $ET_w$.

The single parameter for the non-dimensional CR, i.e., $\alpha_e$, could be analytically obtained by inserting the Priestley-Taylor equation into the Bowen ratio of a wet environment (Szilagyi et al., 2017):


$$\alpha_e = \frac{[\Delta(T_a)+\gamma][e_s(T_{ws})-e_s(T_{dew})]}{\Delta(T_a)\{[e_s(T_{ws})-e_s(T_{dew})]+\gamma[T_{ws}-T_a]\}},$$ (8)

where, $\alpha_e$ must be fall within the theoretical limit of [1, 1+$\gamma$/$\Delta(T_a)$] (Priestley and Taylor, 1972).

**2.2 Data used for $ET_a$ estimation and performance evaluation**

Since Eq. (8) is applicable only in a wet environment, Szilagyi et al. (2017) identified wet locations in a continental area using the fact that the air close to saturation is likely to have high relative humidity (RH) and $T_{ws}$ higher than $T_a$. Thus, $\alpha_e$





values were calculated at locations with RH > 90% and $T_{ws} > T_a + 2$ °C, and their average was assumed to be an unbiased $\alpha_e$ for every location of interest.

However, the spatially constant $\alpha_e$ may not be suitable in a continental area, because the dynamic equilibrium between the atmosphere and the underlying surface is intertwined with partitioning of P into $ET_a$ and runoff (Q). Kim and Chun (2021) analytically linked Eq. (1) with the Turc-Mezentsev equation, and explained the variation of x with climatological aridity

and an implicit land-surface parameter. To satisfy the underpinning independence between P and $R_n$, they reformulated the traditional Budyko equation with $\Phi_0 \equiv ET_w/P$ in lieu of the commonly used aridity index (i.e., $\Phi \equiv ET_p/P$) as:

$$\frac{ET_a}{P} = \frac{ET_w}{P}\left[\frac{1}{1+\left(\frac{ET_w}{P}\right)^n}\right]^{\frac{1}{n}} = \frac{xET_p}{P}\left[\frac{1}{1+\left(\frac{xET_p}{P}\right)^n}\right]^{\frac{1}{n}},$$ (9)

where, n is the land-surface parameter that accounts for factors other than climatic controls affecting the partitioning of P. By dividing Eq. (9) with $\Phi$, it is found that the P partitioning is intertwined with the dimensionless CR as:

$$y = \frac{ET_a}{ET_p} = 2X^2 - X^3 = \left[\frac{x^n}{1+x^n\Phi^n}\right]^{\frac{1}{n}}.$$ (10)

Eq. (10) implicates that the CR needs to be constrained by climatological aridity and surface properties.

When $ET_a$ and P data are available at a sufficient number of river basins, Eq. (10) enables users to estimate x and n. Considering $x_{min} = xET_p/E_{pmax}$, the non-linear Eq. (10) could be simplified by x values from Eq. (10) and corresponding $\Phi$, $ET_p/E_{pmax}$, and n as:

$$\hat{x} = b_0 + b_1 \ln(\Phi) + b_2 \ln(ET_p/E_{pmax}) + b_2 \ln(n),$$ (11)

where, $\hat{x}$ is the climatologically unbiased ratio of $ET_w$ to $ET_p$, and $b_0$, $b_1$, and $b_2$ are the intercept and the regression coefficients, respectively. For ungauged locations where n is unavailable, Eq. (11) could be further approximated only using the climatic variables:

$$\hat{x} = c_0 + c_1 \ln(\Phi) + c_2 \ln(ET_p/E_{pmax}),$$ (12)

where, $c_0$, $c_1$, and $c_2$ are the intercept and the regression coefficients of the approximated equation. Using $\hat{x}$ from Eq. (12), one could estimate $\alpha_e$ in an ungauged location as:

$$\hat{\alpha}_e = \hat{x}\frac{ET_p}{ET_{eq}}$$ (13a)

$$ET_{eq} = \frac{\Delta(T_w)}{\Delta(T_w)+\gamma}\frac{R_n}{\lambda_v}$$ (13b)

where, the estimated $\hat{\alpha}_e$ approximately satisfies the CR and the Budyko equation together, and $ET_{eq}$ is the equilibrium

evapotranspiration (mm d$^{-1}$). Note that P, $ET_p$, $E_{pmax}$, and $ET_{eq}$ within Eqs. (9)-(13) must be on a timescale where the Turc-Mezentsev equation is valid (typically longer than a year), and $\hat{\alpha}_e$ should be bounded with the theoretical limits of [1, $1+\gamma/\Delta(T_a)$].



**2.3 Atmospheric forcing, eddy-covariance, and runoff datasets for application**

We examined the CR combined with the Budyko framework in Australia lying within [10°- 45° S, 113°- 155° E]. The
atmospheric forcing data ($R_n$, $T_a$, $T_{dew}$, and $u_2$) were collected from the advanced ERA5-Land reanalysis archive (Muñoz-Sabater et al., 2021) of the European Centre for Medium-Range Weather Forecasts (https://cds.climate.copernicus.eu; last access on Dec-10/2021). The monthly averages of surface latent and sensible heat fluxes, 2-m air temperature, 2-m dew-point temperature, and 10-m U and V wind speed components at 0.1°×0.1° were downloaded for 1981-2020. $R_n$ was calculated by summing the two heat fluxes, and the 10-m wind speed components were converted to $u_2$ using the logarithmic
vertical profile (Allen et al., 1998).

As a point-scale evaluation reference, monthly latent heat flux observations at the 16 eddy-covariance stations in Table 1 were taken from the FLUXNET2015 archive (https://fluxnet.org/; last access on Jul-1/2021). We chose the flux towers at which 24 or more monthly data with high quality ('LE_F_MDS_QC' > 0.95), and employed the latent heat flux data multiplied by the energy balance closure correction factor. Considering the fine resolution of the ERA5-Land forcing
data, we believed that the resulting CR $ET_a$ estimates could be compared directly with the point-scale observations.

As a basin-scale evaluation reference, we also collected the Australian edition of the Catchment Attributes and Meteorology for Large sample Studies (CAMELS; Fowler et al., 2021) series of datasets (available at https://doi.org/10.1594/PANGAEA.921850; last access on Sep-27/2021). The CAMELS datasets comprise daily time series of 19 hydrometeorological variables at 222 unregulated river basins in Australia. We took P and runoff (Q) data for 1981-
2014 in 71 river basins larger than 500 km² that could contain at least five CR $ET_a$ estimates at 0.1°×0.1°. The basin-scale water balance was approximated by $ET_{wb} \approx \Sigma P - \Sigma Q$, where $ET_{wb}$ is water-balance $ET_a$ at the mean annual scale.

In addition, the SILO P data at 0.01°×0.01° were collected from the Queensland government (https://www.longpaddock.qld.gov.au/silo/gridded-data; last access on Jun-01/2021) together with the Global RUNoff (GRUN) ENSEMBLE data (Ghiggi et al., 2021) (https://doi.org/10.6084/m9.figshare.12794075; last access on Oct-1/2021).
The global Q data were produced at 0.5°×0.5° using a machine-learning algorithm trained by in-situ streamflow observations, and potential errors were reduced by simulations with 21 different sets of atmospheric forcing (Ghiggi et al., 2021). After bilinearly unifying the resolutions of SILO P and GRUN Q datasets, we calculated the mean annual $ET_{wb}$ for 1981-2016 at 0.5°×0.5° over the entire Australian continent.

Against the grid-scale $ET_{wb}$ estimates, predictive performance of the CR method was compared with three $ET_a$
products from a physical, a machine-learning, and a land-surface models. The physical model was the Global Land Evaporation Amsterdam Model (GLEAM) v3.2 (Martens et al., 2017; https://www.gleam.eu; last access on Jun-03/2020) based on the Priestley-Taylor equation constrained by microwave-derived soil moisture, surface temperature, and vegetation optical depth. The machine-learning $ET_a$ product was the FluxCom (http://www.fluxcom.org/; last access Mar-18/2019) that upscaled in-situ observations at 224 eddy-covariance towers using 11 algorithms (Jung et al., 2019). Among the variations of
the FluxCom products, we chose the one forced by the CRUNCEPv8 that has the longest data length from 1950 to 2016. The





land-surface-model-based product was the ERA5-Land monthly $ET_a$ (https://cds.climate.copernicus.eu; last access on Jul-7/2021) simulated by the advanced Hydrology Tiled ECMWF Scheme for Surface Exchanges over Land scheme (Balsamo et al., 2015). We bilinearly unified the different resolutions of the modeled $ET_a$ products to 0.5°×0.5°, and their common period of the modeled $ET_a$ products was 1981-2016.

## 3 Results

### 3.1 Performance of the calibration-free CR in Australia

Figure 1a depicts the spatial distribution of the inverse of aridity index ($\Phi^{-1} = P/ET_p$) that has been traditionally used for climate classification. The mean ratio between SILO P and $ET_p$ for 1981-2014 shows that 81% of the Australian land surfaces were under arid ($\Phi^{-1} < 0.2$) and semi-arid climates ($0.2 < \Phi^{-1} < 0.5$). Semi-humid ($0.5 < \Phi^{-1} < 0.65$) and humid climates ($\Phi^{-1} > 0.65$) were only found in the northern and southeastern coastal areas and the southwestern edge where major cities and agricultural lands have developed. The blue-colored areas in Figure 1a are the locations with RH > 90% and $T_{ws} > T_a + 2°C$, at which the $\alpha_e$ values from Eq. (8) were within 1.15 ± 0.064 (median ± interquartile range). Though the two conditions were satisfied in some mountainous areas in the southeastern part, we excluded them because unexpectedly high $\alpha_e$ values were obtained. The median $\alpha_e = 1.15$ fell within the theoretically acceptable range, and was close to the values found by Ma et al. (2019) and Ma and Szilagyi (2019).

Using $\alpha_e = 1.15$, we synthesized CR $ET_a$ over the entire Australian continent (Figure 1b). The distribution of the mean CR $ET_a$ for 1981-2014 was coherent with that of $\Phi^{-1}$. The mean CR $ET_a$ ranged in 248 ± 99.7 mm a$^{-1}$ and 547 ± 252 mm a$^{-1}$ under arid ($0.05 < \Phi^{-1} < 0.25$) and semi-arid ($0.25 < \Phi^{-1} < 0.50$) climates, respectively. In contrast, CR $ET_a$ in semi-humid ($0.5 < \Phi^{-1} < 0.65$) and humid ($\Phi^{-1} > 0.65$) locations were much higher, being within 913 ± 293 mm a$^{-1}$ and 960 ± 333 mm a$^{-1}$, respectively. Hyper-arid climates ($\Phi^{-1} < 0.05$) were not found in Australia. The continental mean CR $ET_a$ was 486.8 mm a$^{-1}$ for 1981-2012, was about 10% higher than the estimate (439 mm a$^{-1}$) in Zhang et al.'s (2016) global-scale synthesis. The continental average of SILO P for 1981-2014 (473.2 mm a$^{-1}$) was slightly smaller than the mean CR $ET_a$, implicating that the calibration-free CR is likely to overrate $ET_a$.

The overestimation of the CR method was confirmed by comparing the $ET_a$ estimates with the flux observations and the basin-scale $ET_{wb}$ (Figure 2). The percent bias (p-bias) of the $ET_a$ estimates were positive to the two observation sets. The regression slopes between estimated and observed $ET_a$ were below 0.75, tending to overate $ET_a$ increasingly as climate becomes wetter. Despite the high Pearson correlation coefficient (Pearson r), the Nash-Sutcliffe efficiency (NSE) and the root mean square error (RMSE) between CR $ET_a$ and $ET_{wb}$ in the CAMELS basins indicated that the calibration-free CR did not perform as highly as in prior studies (Ma et al., 2021, Ma and Szilagyi, 2019; Ma et al., 2019: Kim et al., 2019).





One may argue that the median $\alpha_e = 1.15$ from the small fractional areas is unlikely representative of the entire Australian continent. Thus, we re-simulated $ET_a$ using the global estimate of $\alpha_e = 1.10$ recently found by Ma et al. (2021). Though the performance measures were improved, the overestimating tendency did not disappear (Figure 3).

### 3.2 The empirical relationship between $\hat{x}$ to climatic variables

Figures 2 and 3 imply that the calibration-free CR with a fixed $\alpha_e$ was unlikely good at closing local water balance
particularly in (semi-)humid river basins. To find climatologically unbiased $\alpha_e$, we first estimated the climatological x and the parameter n of the CAMLES basins using Eq. (10) using the averages of $ET_{wb}$, P, $ET_p$, and $E_{pmax}$ over 1981-2014. Figure 4a-c shows the scatter plots between the resultant x and corresponding $\Phi$, $ET_p/E_{pmax}$, and n values. The The Pearson r between the x and the other three variables was -0.83, -0.49, and 0.44, respectively (significant at 1% level), suggesting that the self-adjustment of $ET_p$ is affected not only by climate conditions, but by land surface properties at least in part.

By regressing the x values with log-transformed $\Phi$, $ET_p/E_{pmax}$ and n, we obtained an empirical relationship that enables to estimate the climatological ratio of $ET_w$ to $ET_p$ as:

$$\hat{x} = 0.964 - 0.206\ln(\Phi) + 0.261\ln\left(ET_p/E_{pmax}\right) + 0.0750\ln(n). \tag{14}$$

The regression coefficients were all significant at 1% level, and the coefficient of determination ($R^2$) was 0.98. The regression equation was further approximated by discarding n from the explanatory variables:

$$\hat{x} = 1.047 - 0.221\ln(\Phi) + 0.251\ln\left(ET_p/E_{pmax}\right). \tag{15}$$

The $R^2$ of the approximated Eq. (15) declined to 0.88. We also found that the simple regression between x and $\Phi$ provided the $R^2$ of 0.84. In other words, though the spatial variation of x could be explained mostly by changes in climatological aridity, heterogeneous land properties might exert non-negligible influences. About 10% of predictability was lost by neglecting the implicit effect of land properties on changes in $\hat{x}$.

Despite the decreased $R^2$, the approximated Eq. (15) performed acceptably in reproducing the x values directly from CR (Figure 4d). The NSE, RMSE, Pearson r, and p-bias between the predicted $\hat{x}$ and the x from CR were 0.88, 0.03, 0.94, and 0.0%, respectively.

### 3.3 Evaluation of annual $ET_a$ and decadal trends against grid-scale water balance

By multiplying $\hat{x}$ to the climatological ratio between $ET_p$ and $ET_{eq}$, we determined $\hat{\alpha}_e$ across the Australian land surfaces.
Figure 5a illustrates the distribution of the resulting $\hat{\alpha}_e$ that varies within $1.08 \pm 0.19$. The median $\alpha_e$ (1.08) was smaller than the Ma et al.'s (2021) global-scale estimate (1.10). The $\hat{\alpha}_e$ values were relatively high in the northwestern and the northern part, whereas they were mostly below the median in the southern and the eastern parts. On 24% of the Australian land surfaces, $\hat{\alpha}_e$ values were unity, implying that they might be below the theoretical lower limit unless bounded.

We re-simulated CR $ET_a$ using the spatially varying $\hat{\alpha}_e$, and found that the overestimating tendency was reduced
considerably (Figure 5b). Under arid and semi-arid climates, the mean CR $ET_a$ ranged within $231 \pm 86.2$ mm a$^{-1}$ and $507 \pm$





247 mm a$^{-1}$ for 1981-2014, while it decreased to 797 ± 406 mm a$^{-1}$ and 806 ± 410 mm a$^{-1}$ in semi-humid and humid regions, respectively. The continental mean $ET_a$ for 1981-2012 declined to 441 mm a$^{-1}$, being practically equal to Zhang et al.'s (2016) estimate, and providing a physically plausible evaporative fraction (93% of P). As expected, the water-balance $ET_{wb}$ in the CAMELS basins were better reproduced by employing the varying $\hat{\alpha}_e$ values, while keeping the point-scale reproducibility at flux tower locations (Figure 6).

Since the empirical Eq. (15) was built by $ET_{wb}$ of the CAMELS basins, one may argue that the evaluation against the same reference would be unfair. Hence, after resampling to 0.5°×0.5°, we compared the CR $ET_a$ estimates against the grid $ET_{wb}$ over the entire Australian continent together with the modeled $ET_a$ by GLEAM, FluxCom, and ERA5-Land. As shown, the CR method with $\alpha_e = 1.15$ overrated the mean annual $ET_a$ for 1981-2016 along the eastern and the northern coastlines (Figure 7b), underperforming the physical, the machine-learning, and the land surface models (Figure 8a). Although the smaller constant $\alpha_e = 1.10$ made the CR method perform better, its predictability was still poorer than the three models and the variation of residuals seemed to be as large as in the simulations with $\alpha_e = 1.15$ (Figure 8b).

When employing the spatially varying $\hat{\alpha}_e$, on the other hand, the same CR formulation could alleviate overestimations along the coastlines (Figure 7c). The varying $\hat{\alpha}_e$ resulted in the CR $ET_a$ estimates agreeing more neatly with the grid $ET_{wb}$, and the variation of residuals was much smaller than in the case of $\alpha_e = 1.10$ (Figure 8c). The CR method with variable $\hat{\alpha}_e$ outperformed the advanced models in reproducing the grid $ET_{wb}$ (Figure 8). Although the referenced $ET_{wb}$ may have some errors associated with upscaling of P and Q observations to the grid scale, our comparative evaluation suggests that discarding the assumption of a fixed $\alpha_e$ could reduce the variation of errors considerably.

## 4 Discussion

### 4.1 Determination of $\alpha_e$ and the Budyko framework

In seven Australian eddy-covariance flux towers, Crago et al. (2022) found that the optimal $\alpha_e$ for the CR of Szilagyi et al. (2017) was 1.35 when predicting daily $ET_a$ in the dimensionless form (i.e., $y = ET_a/ET_p$). However, it should be increased to 1.42, 1.45, 1.47, and 1.50 to simulate dimensional latent heat fluxes at daily, weekly, monthly, and annual timescales, respectively. In Crago and Qualls (2018), the optimal $\alpha_e$ for the kindred linear CR of Crago et al. (2016) varied between 1.00 and 1.43. The prior point-scale experiments have already suggested that a constant $\alpha_e$ is unlikely suitable for the non-dimensional CRs to predict $ET_a$ in Australia.

Evidently, the ratio between the aerodynamic and the radiation components of the Penman equation is affected by the entrainment from the top of the boundary layer (Baldocchi et al., 2016), the dissimilarity between heat and water vapor sources (Assouline et al., 2016), the large-scale synoptic changes (Guo et al., 2015), the horizontal advection of dry air mass (Jury and Tanner, 1975) and among others. More recently, Han et al. (2021) proved the non-linear dependence of $ET_w$ on $ET_{eq}$ using the sigmoid CR of Han and Tian (2018). Yang and Roderick (2019) empirically found that $\alpha_e$ varies with $R_n$ even





over ocean surfaces. The theoretical and empirical evidence is counterintuitive to the constant $\alpha_e$ assumption underpinning the calibration-free CR.

Although Ma et al. (2021) highlighted the global applicability of the calibration-free CR, its performance was remained unknown in most of the Australian land surfaces and in many ungauged basins over the world. Given the spatially diverse climate conditions, assuming a single $\alpha_e$ value across the continental area is questionable. Here, we analytically addressed that the dimensionless CR relates to the Budyko framework describing the long-term water balance simply with the climatological aridity. The Turc-Mezentsev equation enables users to develop an empirical relationship between climate (i.e., $\Phi$ and $ET_p/E_{pmax}$) and the degree of $ET_p$ adjustment (i.e., $\hat{x}$), making the CR method better close local water budgets. The comparative evaluation highlights that the $\alpha_e$ values constrained by diverse climate conditions is likely to make the CR method outperform the advanced physical, machine-learning, and land surface models. Thus, neglecting local P data may not be a good choice when predicting $ET_a$ with the CR method in ungauged areas. It is noteworthy that $\Phi$ was the dominant control of the $\hat{x}$ variation.

While here we addressed the problematic assumption of spatially constant $\alpha_e$, more questions could be raised when employing the polynomial or a kindred CR. For example, the $\alpha_e$ values obtained from the Rome wind function would inherently rely upon an unrealistic assumption that the aerodynamic resistance on a vegetated surface is equivalent to that of open-water surfaces. However, it is unknown if this assumption is practically valid, because the Penman equation formulated with the Rome wind function may result in unrealistically high $ET_p$ even over large wet areas (McMahon et al., 2013). Given the importance of the aerodynamic resistance in modulating surface temperature (Chen et al., 2020), ignoring its temporal variability may become a considerable error source may affect the performance of the CR method sub-annual timescales. Since the steady-state Budyko equation is unlikely to resolve this problem, further improvements are necessary for the CR formulations.

### 4.2 Limitations

We employed the meteorological data different from those used in Ma et al. (2021). The ERA5-Land data are more advanced and produced at a higher resolution than the ERA5 data (Hersbach et al., 2020) by which Ma et al. (2021) predicted $ET_a$ globally. Ma et al. (2021) incorporated remotely sensed albedo and emissivity together with a correction factor when calculating $R_n$, whereas we directly employed the sum of the ERA5-Land latent and sensible heat fluxes. Those input differences, too, may lead to discrepancy in CR $ET_a$ estimates.

The gridded GRUN Q dataset also has some uncertainty sources, though it is the ensemble of many runoff simulations from 21 different atmospheric forcing inputs. In the associated machine-leaning process, some Q observations affected by human activities (e.g., dam regulation and return flows from groundwater abstraction) might not be excluded, potentially disrupting the empirical relationship between atmospheric forcing and natural flows (Ghiggi et al., 2021). In addition, the uncertainty of SILO P might be non-negligible in areas with limited weather stations and in mountainous areas





(Fu et al., 2022). Though we reduced the potential errors in the gridded P and Q datasets by temporal averaging, the grid-
310 scale $ET_{wb}$ estimates should be treated as plausible values rather than exact observations.

## 5 Conclusions

In this work, we showed the calibration-free CR is unlikely to perform well in Australia due at least to the assumption of a constant Priestley-Taylor coefficient. We resolved this problem by linking the CR with the traditional Turc-Mezentsev equation, and drew the following conclusions:

(1) The constant Priestley-Taylor coefficient transferred from fractional wet locations could lead the CR method to poor performance in closing basin-scale water balance. The CR with a constant Priestley-Taylor coefficient seemed to underperform the widely used physical, machine-learning, and land surface models.

   (2) The Budyko framework could provide an additional condition that constrains the degree of $ET_p$ adjustment at the mean annual scale, upscaling the optimal Priestley-Taylor coefficients from gauged to ungauged locations.

(3) The Priestley-Taylor coefficients constrained by diverse climate conditions showed outstanding performance in closing the local water balance over the Australian continent, and the CR method outperformed the other advanced $ET_a$ models.

## Author contributions

DK, MC, and JAC organized this study together. DK built the research framework, simulated $ET_a$ with the CR method, and drafted the manuscript. JAC processed the modeled $ET_a$ datasets and reviewed the draft, and MC actively participated in discussion.

## Competing interests

The authors declare no competing interests.

## 330   Code availability

The Python scripts that implement the CR method are available upon request from the leading author (daeha.kim@jbnu.ac.kr).



## Acknowledgements

This study is supported by the APEC Climate Center. We also acknowledge the financial support of the National Research
Foundation of Korea (NRF) funded by the Korea government (MSIT) (NRF-2019R1A2B5B01070196).

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



**Table 1. List of the chosen FLUXNET sites**

| Site ID | Lon. (°E) | Lat. (°S) | Data period | Site ID | Lon. (°E) | Lat. (°S) | Data period |
|---------|-----------|-----------|-------------|---------|-----------|-----------|-------------|
| AU-ASM  | 133.25    | 22.28     | 2010-2014   | AU-How  | 131.15    | 12.49     | 2001-2014   |
| AU-Cpr  | 140.59    | 34.00     | 2010-2014   | AU-Rig  | 145.58    | 36.65     | 2011-2014   |
| AU-DaP  | 131.32    | 14.06     | 2007-2013   | AU-Stp  | 133.35    | 17.15     | 2008-2014   |
| AU-DaS  | 131.39    | 14.16     | 2008-2014   | AU-TTE  | 133.64    | 22.29     | 2012-2014   |
| AU-Dry  | 132.37    | 15.26     | 2008-2014   | AU-Tum  | 148.15    | 35.66     | 2001-2014   |
| AU-Emr  | 148.47    | 23.86     | 2011-2013   | AU-Wac  | 145.19    | 37.43     | 2005-2008   |
| AU-Fog  | 131.31    | 12.55     | 2006-2008   | AU-Whr  | 145.03    | 36.67     | 2011-2014   |
| AU-Gin  | 115.71    | 31.38     | 2011-2014   | AU-Wom  | 144.09    | 37.42     | 2010-2014   |




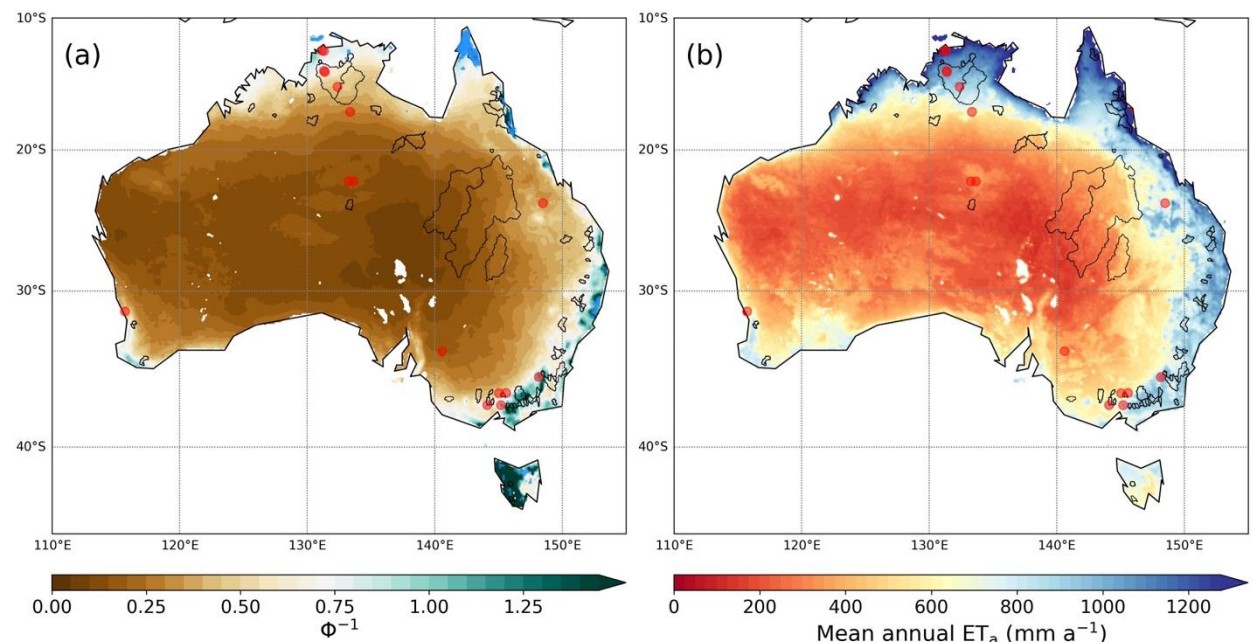

**Figure 1: The spatial distributions of (a) wetness index and (b) mean annual ET$_a$ for 1981-2014 predicted by the calibration-free CR. The red circles and the gray polygons are the chosen flux towers and the CAMELS river basins. The blue-colored areas in (a) indicate the wet cells identified by RH > 90% and T$_{ws}$ > T$_a$ + 2 °C. CR ET$_a$ was simulated at grid cells where the land fraction was**
**larger than 50%.**



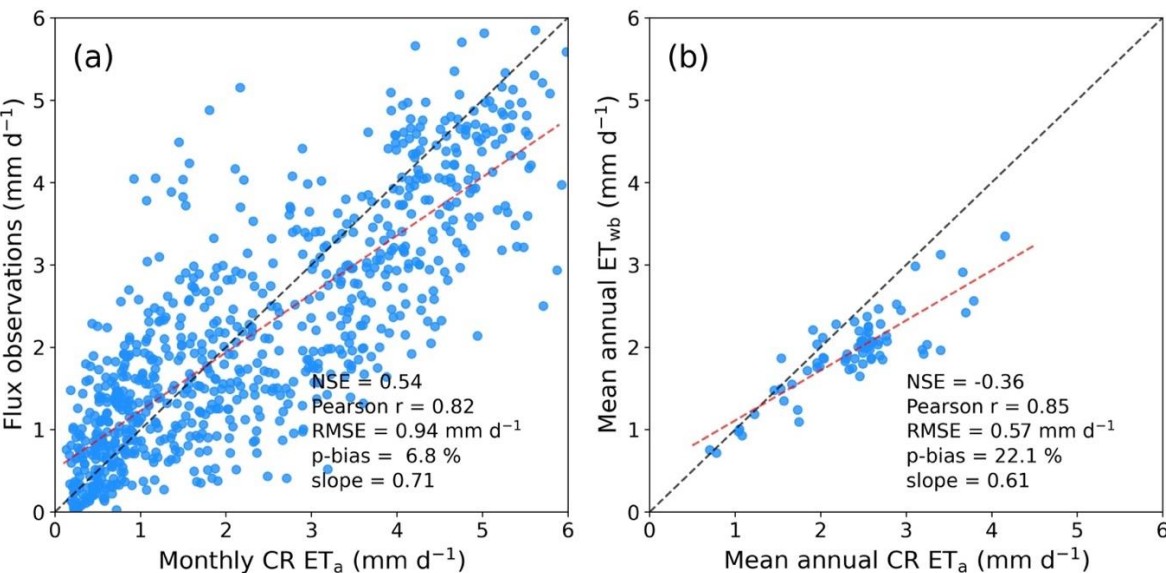

**Figure 2: The 1:1 comparison between the calibration-free CR ETₐ estimates against (a) the monthly FLUXNET2015 observations and (b) the mean annual ETwb at 71 CAMELS for 1981-2014 predicted by the calibration-free CR.**





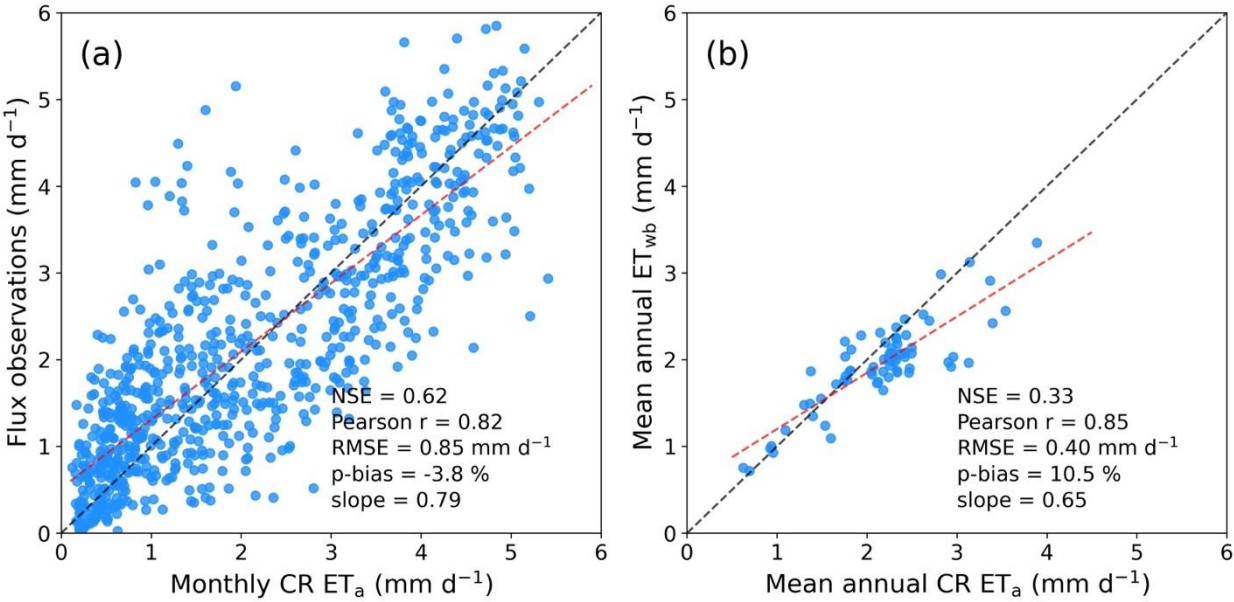


**Figure 3: As in Figure 2, but with $\alpha_e = 1.10$.**

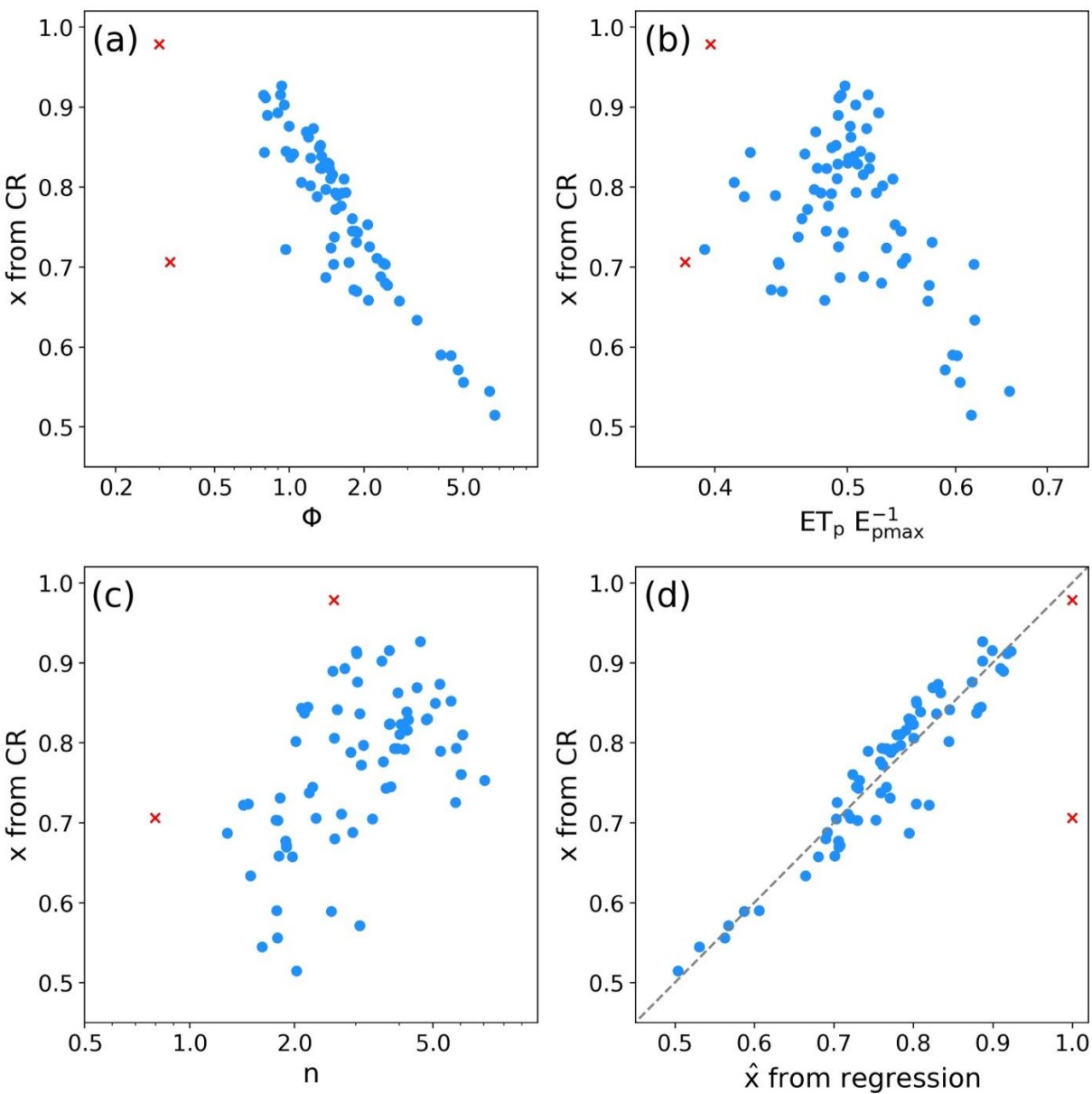

**Figure 4: The scatter plots between the x from the CR with ET$_{wb}$ and corresponding (a) Φ, (b) ET$_p$/E$_{pmax}$, and (c) n values, and (d)**
**the 1:1 plot between the x and the predicted x̂ by Eq. (15). The red x symbols are the outliers excluded from the correlation and**
**the regression analyses.**





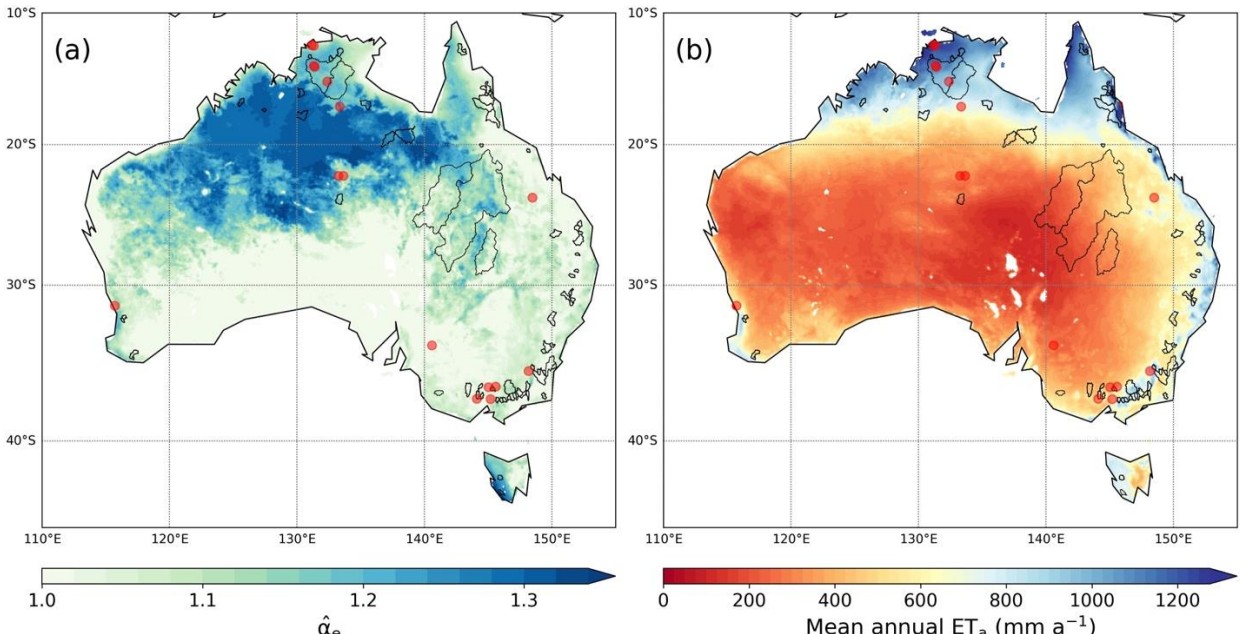

**Figure 5: The distributions of (a) $\hat{\alpha}_e$ values upscaled by the Budyko framework, and (b) the mean annual ET$_a$ predicted by the CR method with $\hat{\alpha}_e$.**



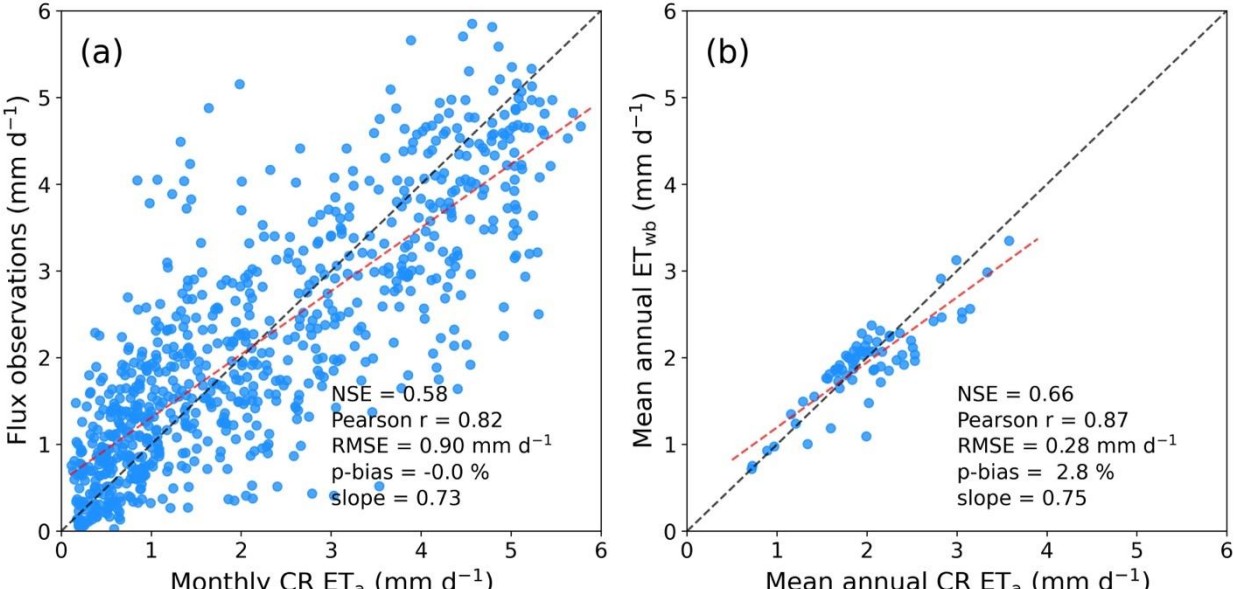

**Figure 6: As in Figure 2, but with $\hat{\alpha}_e$ varying across the land surfaces.**



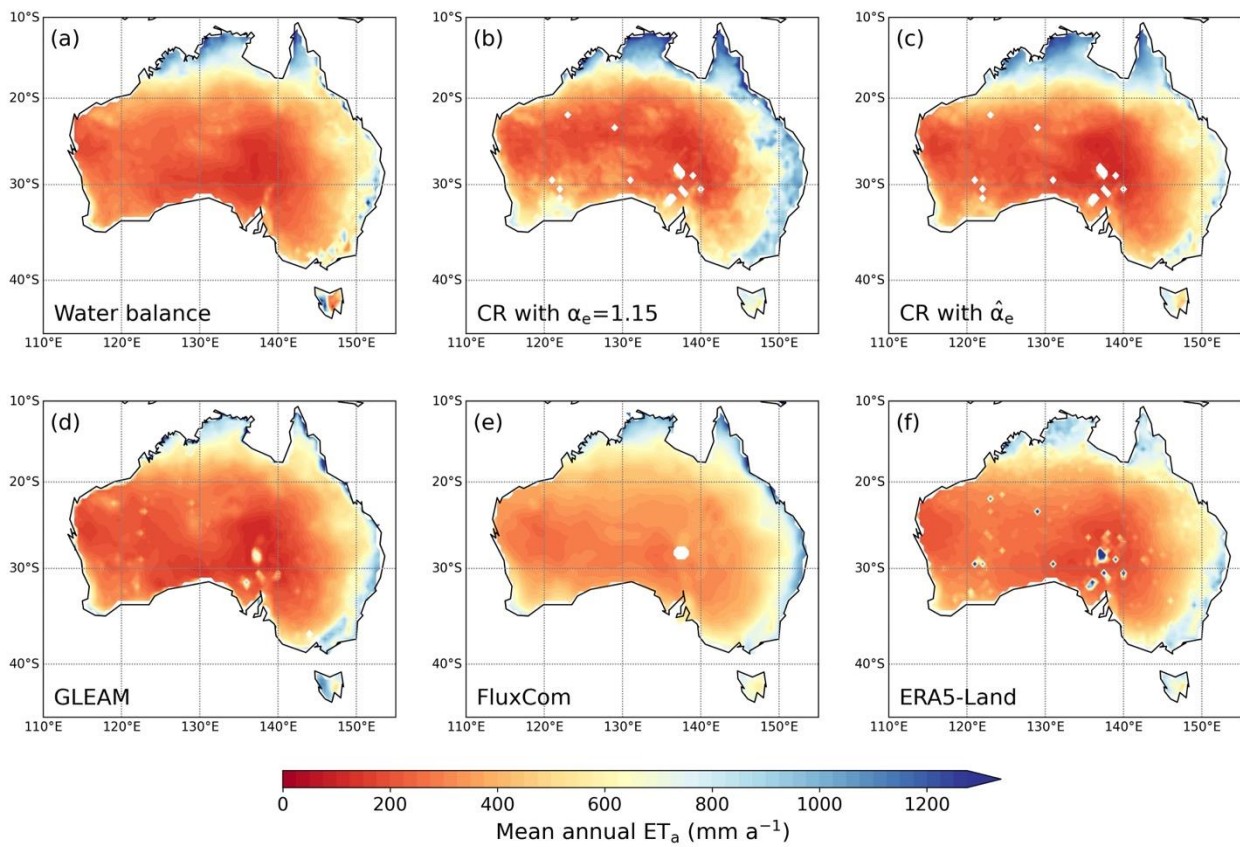

**Figure 7: The distributions of (a) the mean annual water-balance ET$_{wb}$ for 1981-2016 and the predictions by (b) CR with α$_e$ = 1.10, (c) CR with α̂$_e$, (d) GLEAM, (e) FluxCom, and (f) ERA5-Land.**





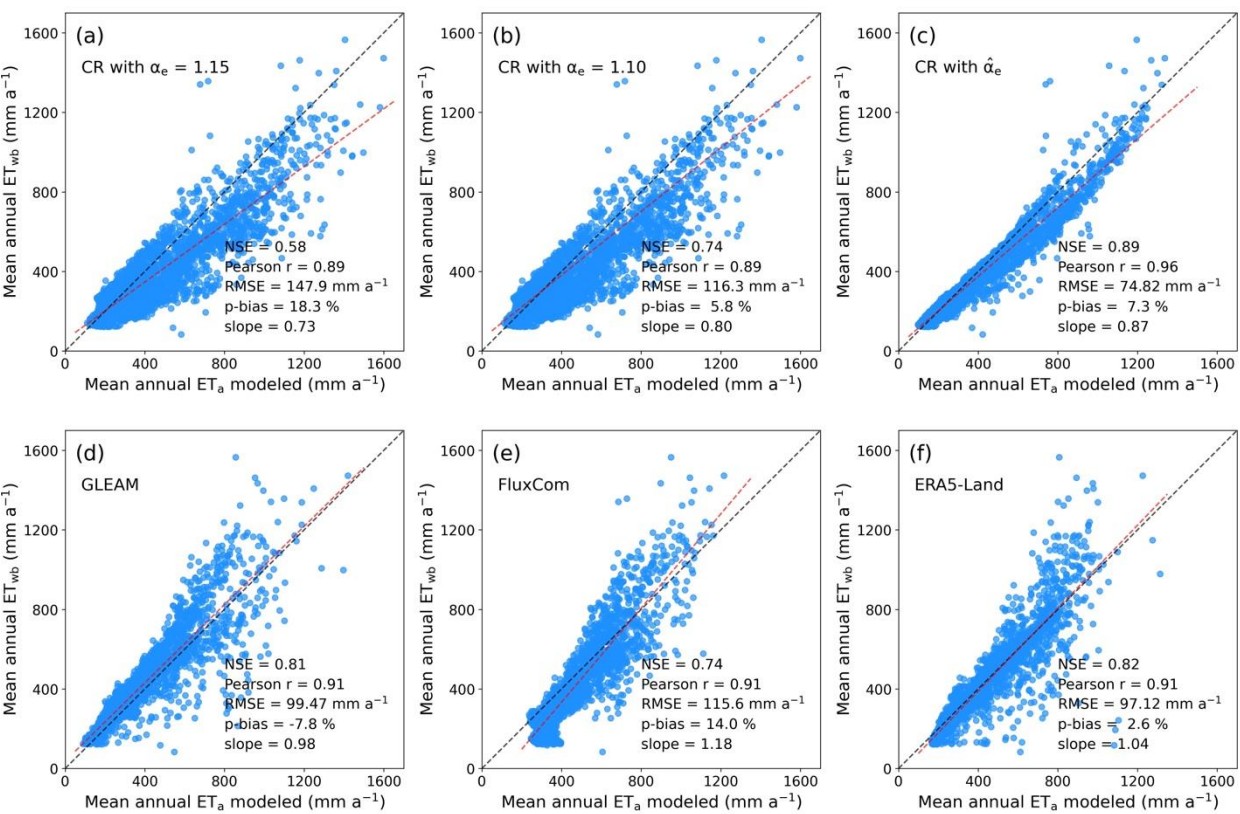


**Figure 8: Scatter plots between the mean annual ET$_{wb}$ for 1981-2016 at 0.5°×0.5° and the predictions by (a) CR with α$_e$ = 1.15, (b) CR with α$_e$ = 1.10, (c) CR with $\hat{α}_e$, (d) GLEAM, (e) FluxCom, and (f) ERA5-Land.**