# Peer review of "Linking the complementary evaporation relationship with the Budyko"

_Hydrology and Earth System Sciences, 2022_

## Author Comment (AC1)

Specific responses to the Referee2's comments

Major comments:
The authors evaluated predictability of the calibration-free complementary relationship (CR) using in-situ flux and runoff observations in Australia, and found that the performance metrics were somewhat lower than shown in prior studies. They potentially attributed the low performance of the calibration-free CR formulation to the unrealistic assumption of the Priestley-Taylor (PT) coefficient, hence proposing an approach that can embed its spatial variation in the CR. To this end, they attempted to connect the polynomial CR with a traditional Budyko equation that has been widely used to describe the surface water balance at a long timescale.

I believe that this paper could help users of the CR method to reliably estimate its single parameter (i.e., the PT coefficient) based on the competition between water and energy balance. I have acknowledged the usefulness of the calibration-free formulation, because its performance could be even better than advanced land surface models in some regions. As the authors stated, the CR methods from the definitive derivation by Brutsaert (2015) have shown outstanding performance in reproducing observed evaporation in many locations over the world, while depending on some parameter calibrations and/or hypotheses that have not been validated. I agree that the assumption of the fixed PT coefficient within a continental-scale area is questionable, because many experimental studies have already found its temporal and spatial variability.

Since the Budyko framework explains the competition between water and energy availability over a land surface, the PT coefficient of the CR method may be better constrained according to overlying climate conditions. The connection to the Budyko framework is likely to lead the CR method to less-biased evaporation estimates, because climate conditions are strongly correlated with vegetation and other land properties. I believe that the findings from this paper have some value for water managers in arid and semi-arid environments, and the topic is interesting and well suited for potential readers of the Hydrology and Earth System Sciences.

Nonetheless, I found some issues in the proposed framework and in the discussion section. Even though they may not be major issues, the authors would need to carefully consider in revision of the manuscript. I would recommend "minor revision" for this manuscript.

→ We greatly appreciate the sound review comments. In this work, we attempted to constrain the Priestley-Taylor (PT) coefficient of the polynomial complementary relationship (CR) with the Budyko framework. The analytical link between the CR and the Turc–Mezentsev equation was already shown by Kim and Chun (2021), and suggests that the self-adjustment of $ET_p$ should be constrained by the aridity index ($\Phi$), i.e., local climate conditions. Since the Budyko equation describes surface water balance that affects the land-atmosphere feedback, it seems obvious that the CR should be in consistency with the Budyko framework.

The calibration-free CR is convenient due to the assumption that the PT coefficients at wet locations could be transferred to any location of interest. However, due to the link between the CR and the Budyko framework (i.e., Eq. 10), the PT coefficient is unlikely to be free of local climate conditions. Here, we highlighted the spatially constant PT coefficient does not guarantee high performance of the polynomial CR in the Australian catchments, whereas the CR constrained by the Budyko framework better reproduced the long-term mean at the catchment and grid scales. Since the constant PT coefficient is not a validated hypothesis by observations, we attempted to test it in the Australian continent where the CR has not been fully examined.

We agree with the comment that our approach is unlikely to improve the calibration-free CR formulation because the empirical relationship between $\hat{x}$ and $\Phi$ and $ET_p/E_{pmax}$ requires some ET data. Hence, we will consider retitling the manuscript to emphasize the additional constraint of the CR method based on the Budyko equation. And, we will also review all the expressions through the manuscript to make our points better presented. Thanks for the positive comments.

Major comments:
Constraining the polynomial CR with the Budyko equation requires sufficient evaporation observations to build the regression relationship between x_hat and the climate variables (Eq. 11). This makes the essential convenience of the calibration-free CR disappear. The proposed framework necessitates any reference evaporation data (e.g., water-balance estimates) to develop the empirical relationship, whereas Szilagyi et al.'s formulation never used reference data. Thus, the proposed framework has pros and cons rather than improving the calibration-free formulation. I think it plays a role in transferring implicit information from gauged to ungauged locations via a solid water-balance equation. On the other hand, Szilagyi et al.'s (2017) approach is applicable only with atmospheric forcing data. Though the constant PT coefficient is a questionable assumption, the two methods have different applicability. So, the title "Improving the calibration-free complementary evaporation principle …" could be somewhat inappropriate. I would suggest retitling it, for example, as "Regionalizing a definitive complementary evaporation relationship by linking with the Budyko framework". I think the new title should imply the authors' intention to regionalize the PT coefficients based on surface water balance explained by the Budyko framework. The introduction should also be reframed accordingly.
→ We agree that our approach has different applicability from that of Szilagyi et al.'s calibration-free CR. We will retitle the manuscript to highlight the Budyko-framework-based constraint for determining the PT coefficient. Yes. It seems to be regionalization of information from gauged to ungauged locations. We will improve the introduction for readers to clearly understand the objective of this paper. Thanks for the good comment.

The section 4.1 seems to overly emphasize limitations of the constant PT coefficient. I would recommend the authors to discuss more about the scientific meaning of the CR-Budyko framework as did in Kim and Chun (2021). This would make this paper more meaningful. Please highlight why CR needs to be constrained even at ungauged locations in the discussion section.
Since the Budyko framework is valid at a long timescale, evaporation observations required for developing Eq. 11 should be sufficiently long. This means that usability of the proposed framework is dependent on regional data availability. Still, Szilagyi et al.'s formulation has better applicability when regional data availability is low. Please add this point in the discussion section, too.
→ We will add the meaning of the Budyko-framework-based constraint in the discussion. Rather than highlighting why the constant PT coefficient is inappropriate, it would be better to explain the physical and mathematical role of the PT coefficient in the CR, and why it should be additionally constrained by the Budyko framework. We will improve the discussion section accordingly. Thanks for the suggestion.

Some grammatical errors and typos are still in the manuscript. Please re-read the manuscript carefully, and correct them including the below technical errors I found.

→ In hindsight, we too found several mistakes from the manuscript. In revision, the typos and wrong expressions will be corrected with a native English speaker.

Some technical errors:
Abstract:
(L12) convenient → highly utilizable
→ We will revise the abstract in accordance with the revised introduction and discussion.
(L14) three advanced ETa models: In Abstract, no explicit list was found. The authors should more explicitly list these three ETa models.
→ We will add the explicit names of the models in the revised abstract.

1 Introduction:
(L39) had found → had been found
→ This is grammatically wrong. We will correct it.

2 Methodology and data:
(L154) The authors may misuse a dash (-) with an en dash (–). Generally, an en dash can replace "to". The authors need to replace the misused dash with an en dash. These replacements should be made throughout the manuscript.
→ The punctuations will be entirely reviewed and corrected.

(L163) ('LE_F_MDS_QC' > 0.95) The authors need to provide more description of this for the potential readers.
→ 'LE_F_MDS_QC' is a name of the columns of the FLUXNET2015 that indicates the quality of data. We will add the clear description in revision.

(L180) a land-surface models → a land-surface model
→ We will review all the expressions with a native English speaker.

3 Results:
(L222–223) The Pearson r between the x and the other three variables was → The Pearson r values between the x and the other three variables were (One "The" must be removed, "values" can be added after "r, and "was" should be changed into "were")
→ We will entirely review the manuscript and correct wrong expressions.

5 Conclusions:
No explicit conclusion was found in this section. The author may change the section title into 5 Summary or add some of conclusions such as recommendations for future research or practical applications.
→ As we did before in many papers, we here provided three points from our analysis and they are concise conclusions of this work. If this short conclusion section is uncomfortable, we will combine the conclusion and the discussion sections.

---

## Author Comment (AC2)

**Specific responses to the Referee1's comments**

I am pleased to review the paper titled "Improving the calibration-free complementary evaporation principle by linking with the Budyko framework". This paper focused on predicting terrestrial evapotranspiration. This method is interesting for calibration-free process. The manuscript is generally well written.

 $\rightarrow$  Thanks for the positive comments. We will revise the manuscript to consider the sound comments of the Referee2.

**Major comments**

There is not enough explanation regarding the model procedure.

 $\rightarrow$  For the CR method, Eq. (1a) is the only equation to estimate ETa. If ETw, ETp, and Epmax are available, then x and xmin could be identified and the y value is obtained. By multiplying ETp to y, ETa is simply calculated. Hence, there is no complex modeling procedures for this method. This simplicity is a great merit, but how to calculate ETw is a difficult problem. Oftentimes, ETw is calculated with the Priestley-Taylor (PT) equation, but how one determines the PT coefficient without reference ETa data is the focus of this study.

**Minor comments**

Please add the difference between the Szilagyi method and the previous method of calibration-free.

 $\rightarrow$  The Szilagyi et al.'s (2017) method is the calibration-free CR. And, we did use the same method, but propose an approach to determining its single parameter (i.e., the PT coefficient) by linking it with the Budyko framework. We will more clearly explain this in revision.

**Eq 1b Why do you choose the min-max scaling?**

→ This min-max scaling is physically essential, because it rescales the range of x from [ $x_{min}$ , 1] to [0, 1]. This rescaling is to correct the implausible boundary condition used in Brutsaert (2015). In Brutseart (2015), the minimum value of  $x = ET_w/ET_p$  was assumed to be zero; however, Crago et al. (2016) argued that  $ET_p$  cannot be infinite in reality or  $ET_w$  is unlikely to be zero over a land surface. Hence, they mended this problem by introducing the maximum  $ET_p$  (i.e.,  $E_{pmax}$ ). If there is no water on a surface,  $ET_p$  should be adjusted to the sensible heat flux. In this case, x is  $x_{min}$ , hence it ranges between  $x_{min}$  (no water) and 1 (ample water). The the min-max rescaling of x provides a convenience to develop Eq. 1a by setting the range of the independent variable within [0, 1].

**References**

Brutsaert, W. (2015), A generalized complementary principle with physical constraints for land-surface evaporation, Water Resour. Res., 51, 8087–8093.

Crago, R., Szilagyi, J., Qualls, R., and Huntington, J. (2016), Rescaling the complementary relationship for land surface evaporation, Water Resour. Res., 52, 8461–8471.

---

## Author Response (AR1)

Dear Dr. Teuling:

We greatly appreciate your editing efforts for handling our manuscript, and thank the two anonymous reviewers for their constructive comments.

We fully considered the given comments in revision. The manuscript was retitled to "Linking the complementary evaporation relationship with the Budyko framework for ungauged areas in Australia" to consider the major comment of Reviewer 1. In the discussion section, we emphasized the physical meaning of the combined CR-Budyko framework, and the remaining issues and caveats were summarized separately. The grammatical errors, typos, and wrong punctuations were checked thoroughly. We note that the point flux data for evaluation were changed from monthly to annual means, because the stationary Budyko equation is valid at an annual or longer timescale. Even after this change, the CR combined with the Budyko framework produced less biased annual ETa than when using Szilagyi et al.'s (2017) original formulation.

We believe that readability of our manuscript is improved by the sound comments from the two reviewers. Once again, we greatly appreciate your editing efforts, and our specific responses to review comments are as follows.

Sincerely,

Jong Ahn Chun
Corresponding author

Specific responses to the Referee 1's comments

Major comments
There is not enough explanation regarding the model procedure.
→ For the CR method, Eq. (1a) is the only equation to estimate $ET_a$. If $ET_w$, $ET_p$, and $E_{pmax}$ are available, then x and $x_{min}$ could be identified and the y value is obtained. By multiplying $ET_p$ to y, $ET_a$ is simply calculated. Hence, there are no complex modeling procedures for this method. This simplicity is a great merit, but how to calculate $ET_w$ is a difficult problem. Oftentimes, $ET_w$ is calculated with the Priestley-Taylor (PT) equation, but how one determines the PT coefficient without reference $ET_a$ data is the focus of this study. For this purpose, we used the analytical connection between the CR and the Budyko framework to constrain the PT coefficient with local climate conditions. In Line 88-129, we described how the CR method previously used, and in Line 131-164, we addressed how to regionalize the PT coefficients from gauged to ungauged locations. This revised version would  more clearly describe the CR, its analytical link to the Turc-Mezentsev equation, and the regression analysis for regionalization.

Minor comments
Please add the difference between the Szilagyi method and the previous method of calibration-free.
→ Szilagyi et al.'s (2017) formulation uses a constant PT coefficient from wet locations. We did use the same equation, but propose an approach to determining its single parameter (i.e., the PT coefficient) by linking it with the Budyko framework. Section 2.2 describes the difference between Szilagyi et al.'s approach and this study (L131-164).

Eq 1b  Why do you choose the min-max scaling?
→ This min-max scaling is physically essential, because it rescales the range of x from [$x_{min}$, 1] to [0, 1]. This rescaling is to correct the implausible boundary condition given by Brutsaert (2015). In Brutseart (2015), the minimum value of $x = ET_w/ET_p$ was assumed to be zero; however, Crago et al. (2016) argued that $ET_p$ cannot be infinite or $ET_w$ is unlikely to be zero over a land surface in reality. Hence, they mended this problem by introducing the maximum $ET_p$ (i.e., $E_{pmax}$). If there is no water on a surface, $ET_p$ should be adjusted to the maximum level ($E_{pmax}$) because all the available radiation would be transformed to the sensible heat flux. In this case, x should reach $x_{min}$, hence it ranges between $x_{min}$ (no water) and 1 (ample water). The min-max rescaling of x leads to Eq. 1a by rescaling the range of the independent variable from [$x_{min}$, 1] to [0, 1].

References

Brutsaert, W. (2015), A generalized complementary principle with physical constraints for land-surface evaporation, Water Resour. Res., 51, 8087– 8093.
Crago, R., Szilagyi, J., Qualls, R., and Huntington, J. (2016), Rescaling the complementary relationship for land surface evaporation, Water Resour. Res., 52, 8461– 8471.

Specific responses to the Referee 2's comments

The authors evaluated predictability of the calibration-free complementary relationship (CR) using in-situ flux and runoff observations in Australia, and found that the performance metrics were somewhat lower than shown in prior studies. They potentially attributed the low performance of the calibration-free CR formulation to the unrealistic assumption of the Priestley-Taylor (PT) coefficient, hence proposing an approach that can embed its spatial variation in the CR. To this end, they attempted to connect the polynomial CR with a traditional Budyko equation that has been widely used to describe the surface water balance at a long timescale. I believe that this paper could help users of the CR method to reliably estimate its single parameter (i.e., the PT coefficient) based on the competition between water and energy balance. I have acknowledged the usefulness of the calibration-free formulation, because its performance could be even better than advanced land surface models in some regions. As the authors stated, the CR methods from the definitive derivation by Brutsaert (2015) have shown outstanding performance in reproducing observed evaporation in many locations over the world, while depending on some parameter calibrations and/or hypotheses that have not been validated. I agree that the assumption of the fixed PT coefficient within a continental-scale area is questionable, because many experimental studies have already found its temporal and spatial variability.

Since the Budyko framework explains the competition between water and energy availability over a land surface, the PT coefficient of the CR method may be better constrained according to overlying climate conditions. The connection to the Budyko framework is likely to lead the CR method to less-biased evaporation estimates, because climate conditions are strongly correlated with vegetation and other land properties. I believe that the findings from this paper have some value for water managers in arid and semi-arid environments, and the topic is interesting and well suited for potential readers of the Hydrology and Earth System Sciences.

Nonetheless, I found some issues in the proposed framework and in the discussion section. Even though they may not be major issues, the authors would need to carefully consider in revision of the manuscript. I would recommend "minor revision" for this manuscript.

→ We greatly appreciate the sound review comments. In this work, we attempted to constrain the Priestley-Taylor (PT) coefficient of the polynomial complementary relationship (CR) with the Budyko framework. The analytical link between the CR and the Turc–Mezentsev equation was already shown by Kim and Chun (2021), and suggests that the self-adjustment of $ET_p$ should be constrained by the aridity index ($\Phi$), i.e., local climate conditions. Since the Budyko equation describes surface water balance that affects the land-atmosphere feedback, it seems obvious that the CR should be in consistency with the Budyko framework.

The calibration-free CR is convenient due to the assumption that the PT coefficients at wet locations could be transferred to any location of interest. However, due to the link between the CR and the Budyko framework (i.e., Eq. 10), the PT coefficient is unlikely to be free of local climate conditions. Here, we highlighted the spatially constant PT coefficient does not guarantee high performance of the polynomial CR in the Australian catchments, whereas the CR constrained by the Budyko framework better reproduced the long-term mean at the catchment and grid scales. Since the constant PT coefficient is not a validated hypothesis by observations, we attempted to test it in the Australian continent where the CR has not been fully examined.

We agree with the comment that our approach is unlikely to improve the calibration-free CR formulation because the empirical relationship between x and $\Phi$ and $ET_p/E_{pmax}$ requires some ET data. Hence, we retitled the manuscript as "Linking the complementary evaporation relationship with the Budyko framework for ungauged areas in Australia". This would better indicate what we intended and where we studied.
In revision, we compared CR ETa with the annual flux observations instead of the previous monthly observations, because the stationary Budyko equation is not valid at a sub-annual timescale. This change would present the performance of the CR-Budyko framework more properly.

Major comments:
Constraining the polynomial CR with the Budyko equation requires sufficient evaporation observations to build the regression relationship between x_hat and the climate variables (Eq. 11). This makes the essential convenience of the calibration-free CR disappear. The proposed framework necessitates any reference evaporation data (e.g., water-balance estimates) to develop the empirical relationship, whereas Szilagyi et al.'s formulation never used reference data. Thus, the proposed framework has pros and cons rather than improving the calibration-free formulation. I think it plays a role in transferring implicit information from gauged to ungauged locations via a solid water-balance equation. On the other hand, Szilagyi et al.'s (2017) approach is applicable only with atmospheric forcing data. Though the constant PT coefficient is a questionable assumption, the two methods have different applicability. So, the title "Improving the calibration-free complementary evaporation principle …" could be somewhat inappropriate. I would suggest retitling it, for example, as "Regionalizing a definitive complementary evaporation relationship by linking with the Budyko framework". I think the new title should imply the authors' intention to regionalize the PT coefficients based on surface water balance explained by the Budyko framework. The introduction should also be reframed accordingly.
→ We agree that our approach has different applicability from that of Szilagyi et al.'s calibration-free CR. We hence retitled the manuscript as "Linking the complementary evaporation relationship with the Budyko framework for ungauged areas in Australia". This title would more clearly indicate what we attempted and where we studied.

The section 4.1 seems to overly emphasize limitations of the constant PT coefficient. I would recommend the authors to discuss more about the scientific meaning of the CR-Budyko framework as did in Kim and Chun (2021). This would make this paper more meaningful. Please highlight why CR needs to be constrained even at ungauged locations in the discussion section.
Since the Budyko framework is valid at a long timescale, evaporation observations required for developing Eq. 11 should be sufficiently long. This means that usability of the proposed framework is dependent on regional data availability. Still, Szilagyi et al.'s formulation has better applicability when regional data availability is low. Please add this point in the discussion section, too.
→ In the revised version, we emphasized the physical meaning of the CR-Budyko framework. Since ETa plays a pivotal role in the water and the energy balance simultaneously the partitioning of net radiation cannot be independent of the partitioning of precipitation. The Budyko framework fills this gap for the CR method by reflecting local

climates in the PT coefficient. We included those points in Section 4.1 (Line 282-287 and 298-305).
The remaining issues and caveats are separately provided in Section 4.2, and they include the scale dependence of the PT coefficients and the aerodynamic resistance. We also include the stationary Budyko equation may not be a solution to problems at sub-annual timescales (L313-323).

Some grammatical errors and typos are still in the manuscript. Please re-read the manuscript carefully, and correct them including the below technical errors I found.
→ We checked the grammatical errors and typos from the beginning to the end. We believe that the readability is considerably improved.

Some technical errors:
Abstract:
(L12) convenient → highly utilizable
→ We corrected as advised.
(L14) three advanced ETa models: In Abstract, no explicit list was found. The authors should more explicitly list these three ETa models.
→ We corrected this expression to "sophisticated physical, machine-learning, and land surface models" for readers to recognize the categories of the chosen models. Since their full names are long, directly listing them seemed to be redundant in the short paragraph. The point in this sentence is the CR method with a constant PT coefficient can underperform commonly-used models.

1 Introduction:
(L39) had found → had been found
→ We corrected the sentence (L37).

2 Methodology and data:
(L154) The authors may misuse a dash (-) with an en dash (–). Generally, an en dash can replace "to". The authors need to replace the misused dash with an en dash. These replacements should be made throughout the manuscript.
→ The punctuations were reviewed and corrected throughout the manuscript.

(L163) ('LE_F_MDS_QC' > 0.95) The authors need to provide more description of this for the potential readers.
→ 'LE_F_MDS_QC' is a name of the columns of the FLUXNET2015 that indicates the quality of data. We added the expression "quality measure" in Line 83.

(L180) a land-surface models → a land-surface model
→ The grammatical error was corrected (L193).

3 Results:
(L222–223) The Pearson r between the x and the other three variables was → The Pearson r values between the x and the other three variables were (One "The" must be removed, "values" can be added after "r, and "was" should be changed into "were")

→ The grammatical errors were corrected  (L242).

5 Conclusions:
No explicit conclusion was found in this section. The author may change the section title into 5 Summary or add some of conclusions such as recommendations for future research or practical applications.
→ The remaining issues are briefly addressed in Section 4.2, and we retitled Section 5 as "Summary".

References

Brutsaert, W. (2015), A generalized complementary principle with physical constraints for land-surface evaporation, Water Resour. Res., 51, 8087– 8093.

Kim, D., and Chun, J. A. (2021), Revisiting a two-parameter Budyko equation with the complementary evaporation principle for proper consideration of surface energy balance. Water Resources Research, 57, e2021WR030838. https://doi.org/10.1029/2021WR030838